# Impact of rapid near-patient STI testing on service delivery outcomes in an integrated sexual health service in the United Kingdom: a controlled interrupted time series study

Scott R Walter [ID],[1,2] Joni Jackson,[1,2] Gareth Myring,[1,2] Maria Theresa Redaniel,[1,2] Ruta Margelyte [ID],[1,2] Rebecca Gardiner,[3,4] Michael D Clarke,[3] Megan Crofts,[3] Hugh McLeod,[1,2] William Hollingworth,[1,2] David Phillips,[5] Peter Muir,[6,7] Jonathan Steer,[6] Jonathan Turner,[6] Paddy J Horner,[2,3,7] Frank De Vocht[1,2]

PJH and FDV are joint first authors.

For numbered affiliations see end of article.

**Correspondence to**
Dr Scott R Walter;
scott.walter@bristol.ac.uk

## ABSTRACT

**Objectives** To evaluate the impact of a new clinic-based rapid sexually transmitted infection testing, diagnosis and treatment service on healthcare delivery and resource needs in an integrated sexual health service.

**Design** Controlled interrupted time series study.

**Setting** Two integrated sexual health services (SHS) in UK: Unity Sexual Health in Bristol, UK (intervention site) and Croydon Sexual Health in London (control site).

**Participants** Electronic patient records for all 58 418 attendances during the period 1 year before and 1 year after the intervention.

**Intervention** Introduction of an in-clinic rapid testing system for gonorrhoea and chlamydia in combination with revised treatment pathways.

**Outcome measures** Time-to-test notification, staff capacity, cost per episode of care and overall service costs. We also assessed rates of gonorrhoea culture swabs, follow-up attendances and examinations.

**Results** Time-to-notification and the rate of gonorrhoea swabs significantly decreased following implementation of the new system. There was no evidence of change in follow-up visits or examination rates for patients seen in clinic related to the new system. Staff capacity in clinics appeared to be maintained across the study period. Overall, the number of episodes per week was unchanged in the intervention site, and the mean cost per episode decreased by 7.5% (95% CI 5.7% to 9.3%).

**Conclusions** The clear improvement in time-to-notification, while maintaining activity at a lower overall cost, suggests that the implementation of clinic-based testing had the intended impact, which bolsters the case for more widespread rollout in sexual health services.

## INTRODUCTION

Sexually transmitted infection (STI) diagnoses are increasing in England with more than a 10% increase in new infections between 2016 and 2019.[1] Over the same period, a 19.2% increase in total consultations

### STRENGTHS AND LIMITATIONS OF THIS STUDY

⇒ We used controlled interrupted time series models with confounder adjustment to estimate the effect of the intervention distinct from any background changes and independent of other time-varying factors.

⇒ Model validity was bolstered by using a relatively long time series with good temporal resolution.

⇒ Data from both the main and control sites were derived from the same electronic patient record system.

⇒ There was a general consensus between main and sensitivity analyses.

⇒ Our study was limited by being non-randomised, having only one control site and the follow-up period for female patients being truncated by the impact of the COVID-19 pandemic.

at sexual health services (SHS) was reported in England.[2] Open-access SHS providing rapid treatment and partner notification can reduce the risk of STI complications and infection spread.[3–5] Public Health England (now UK Health Security Agency (UKHSA)) recommends that local SHSs need to be available to both the general population and groups with greater sexual health needs.[3] Nevertheless, the central government's public health grant, including SHS funding, has steadily decreased since 2015.[6 7] Despite diminishing resources, continued provision of SHSs has been achieved through increased efficiencies at clinic-based services and introduction of online services.[8 9]

Another approach to improving efficiency while ensuring quality could be the introduction of near-patient testing (NPT) for chlamydia and gonorrhoea. That is, testing

where samples are taken at the time of consultation and results returned within a short timeframe (immediately or within hours). Potential benefits include earlier diagnosis and treatment, reduced risk of sequelae and onward transmission and reduction in unnecessary treatments as well as reduced costs and clinician time due to reduction in the need for gonorrhoea cultures (GCs), examinations and follow-up visits.[10–12] Although modelling studies suggest that NPT can be cost-effective, this remains to be demonstrated in practice.[10–14] Research also suggests that reduced waiting times for STI test results may enhance patient acceptability[15 16] and increase testing uptake.[17 18] Importantly, patients have expressed preferences for earlier provision of results[19] due to the stress of waiting.[20]

In November 2018, Unity Sexual Health (hereafter *the intervention site*), a UK specialist integrated SHS, implemented a rapid nucleic acid amplification (NAAT) STI testing, diagnosis and treatment service for chlamydia and gonorrhoea, using the Hologic 'Panther' diagnostic platform in a clinic-based satellite laboratory.[21] It can deliver results in 3.5 hours by eliminating sample batching and transit times associated with microbiology laboratory testing. Integrated SHS provide the full range of contraception services in addition to STI and blood borne virus testing, treatment and management, and health promotion and prevention.[22]

We used a quantitative approach to evaluate the impact of the new rapid testing process on service delivery and resource needs of the intervention site.

## METHODS
### Setting and design
The intervention site is a provider of integrated SHSs in the Bristol area of the United Kingdom, with about 40 000 attendances annually. In addition to in-clinic services, self-testing kits for chlamydia, gonorrhoea, syphilis and HIV ordered online by patients are provided by post. This postal testing kit service was provided by the intervention site for asymptomatic patients through its dedicated website and used the same NAAT testing platform as the rapid STI service. This was in place prior to the intervention and was increasingly used throughout the study period.

This study is a quasi-experimental, controlled interrupted time series (CITS) design that used routinely collected electronic patient record (EPR) data. The intervention time points were defined differently for males and females: rapid STI testing was introduced on 12 November 2018 for males and 29 May 2019 for females.

### Rapid STI service model
Eligibility criteria and treatment pathways differed for males and females. A graphical overview of each pathway is provided in the supplement (online supplemental figures S1 and S2) with preintervention pathway included for reference. Additional changes were made to the SHS

related to staff capacity. Rapid STI asymptomatic consultations were reduced to 15 min, while the number of allocated patients per staff member for the walk-in clinic remained the same.

### Rapid STI testing
The collection, processing and analysis of specimens with the Aptima Combo 2 (Hologic) NAAT at the intervention site, which detects both *Chlamydia trachomatis* and *Neisseria gonorrhoeae* and the Aptima TV *Trichomonas vaginalis* NAAT, followed the manufacturer's instructions and national guidelines. Quality control measures were the same as those in the central UKHSA South West Regional Laboratory and complied with national standards. The testing was undertaken by a dedicated technician employed by UKHSA experienced in using the Hologic Panther platform (further details in online supplemental file 1).

### Male patients
Male patients were eligible for the rapid STI pathway if they were asymptomatic or had urethritis symptoms. If asymptomatic, a brief history was taken prior to patient self-sampling for chlamydia and gonorrhoea and taking blood tests for HIV and syphilis. Men who have sex with men (MSM) were referred to a health adviser for health promotion, including discussion about testing for HIV and other STIs, and safer sex practices. Symptomatic males were asked to return 4 hours later when NAAT results were available. If positive, they received infection-specific treatment; if negative, a urethral smear was undertaken to diagnose non-gonococcal urethritis. Contacts of patients with gonorrhoea or chlamydia outside a 2-week window were treated if NAAT positive. Swabs for gonococcal culture and sensitivities were only taken after a NAAT-positive result for gonorrhoea or if gonococcal treatment was administered prior to the NAAT result.

### Female patients
Female asymptomatic patients without contraception needs were eligible for the rapid drop-off service. Females with abnormal vaginal discharge, not requiring bimanual or speculum examination to exclude pathology, self-swabbed and were treated on the results of microscopy and clinical findings at the time of visit and informed that chlamydia and gonorrhoea NAAT test results would be available within 48 hours. They were termed symptomatic. For contraceptive needs, a clinical consultation was necessary to determine the need for examination. Trichomonas vaginalis (TV) culture was replaced with a more sensitive TV NAAT,[23] also available within 48 hours. A gonococcal culture swab was only taken after a NAAT-positive result for gonorrhoea or if gonococcal treatment was administered prior to NAAT result.

### Control site
Croydon Sexual Health, a similar integrated SHS in South London, was used as the control site to account for background changes unrelated to the intervention. This

site has similar patient throughput (about 32 000 annual attendances) and uses the same EPR system.

## Data

Fully anonymised individual patient data extracted from the intervention and control site EPR systems[23] comprised demographic information, sexual behaviour, mode of presentation and attendances to the clinic, diagnostic testing and treatment. Analyses were based on a census of attendance level records.

Time-to-notification was defined from the text message notification system.[24] This included text message type for identifying test result messages, time stamps and anonymised patient identifiers. The number of NAAT postal testing kits was extracted from the intervention site's records, while the control site did not implement these until after the study period.

Prior to analysis, data were checked for duplicates, implausible values and missingness. Individual variables were combined to generate indicator variables for complex cases, MSM, examinations and ethnic minority status. All time-related variables were derived from the date and time of each attendance.

For analysis, data were aggregated at weekly level over a 2-year period centred at the intervention. For females, data were excluded from the first UK COVID-19-related lockdown (23 March 2020) due to changes in outcomes that could not be adequately accounted for in models. The study period for males was from 13 November 2017 to 10 November 2019, and for females from 28 May 2018 to 22 March 2020.

## Statistical analysis

Their main study outcomes are detailed in table 1. CITS models within a generalised linear modelling framework were applied to each outcome separately for males and females: 10 models in total. *Time* was modelled as linear using consecutively numbered weeks, with *time*=0 at the intervention point. A binary variable (*period*) representing pre-intervention and post-intervention periods was defined by the respective male and female intervention dates.

GC swabs per consultation, follow-up attendances per care episode, examinations per symptomatic attendance and staff capacity were modelled as rates assuming a negative binomial distribution. These models generate rate ratios, presented as percentage changes. For time-to-notification, a normal distribution was assumed and results were presented as differences in median time (days). This represents absolute measure of time, including weekends as opposed to working days only.

The main variables in the models were *time*, *period* and *site* (intervention vs control) along with all two-way and three-way interactions, as per a CITS approach for estimating both a step change and slope change.[25 26] Two key terms in the models represent intervention-related changes over and above any control-site changes. The *period × site* interaction captures a differential step change

**Table 1** Definitions of main study outcomes for weekly aggregated data

| Outcome measure | Definition |
|---|---|
| 1. Rate of gonorrhoea culture swabs per consultation | Numerator: the number of GC swabs, urethral for male and cervical for female<br>Denominator: the number of consultations where these were defined as attendances for new, rebooked or walk-in patients |
| 2. Time-to-notification | Median time from sample collection until the patient was notified of the test result via text message |
| 3. Rate of examinations per symptomatic attendance | Numerator: the number of examinations of any type. This was based on a combination of variables used to record information about examinations (online supplemental table S1)<br>Denominator: all attendances where the patient was recorded as being symptomatic |
| 4. Rate of follow-up attendances per episode of care | Numerator: the number of follow-up attendances occurring within 30 days of an initial consultation<br>Denominator: the number of episodes involving at least one consultation |
| 5. Staff capacity: rate of patients seen per 4 hour clinic | Numerator: number of patient consultations (any new, rebooked, walk-in or follow-up attendance)<br>Denominator: number staff available for 4 hour clinics |

GC, gonorrhoea culture.

for the intervention site compared with control site. While the three-way interaction term *time × period × site* captures different degrees of pre–post trend change for the intervention site compared with control site (online supplemental figure S3).

Additional covariates were included in the models: proportions of complex patients, symptomatic patients and patients from an ethnic minority, plus mean patient age and calendar month. Since models of examination rate only analysed symptomatic patients, the proportion of symptomatic patients was excluded as a covariate. The proportion of MSM was only included in models for men. Complex cases were defined differently for males and females (see online supplement). This is based on the definition used by Mohiuddin *et al*[12] designed to identify patients requiring longer and/or more involved consultations.

Data for staff capacity were only available for the intervention site and were modelled as an uncontrolled interrupted time series spanning the duration of available denominator data: 1 January 2018 to 22 December 2019. The denominator could not be separated by sex, so this

outcome was analysed for females and males combined, allowing two change points as per the respective intervention dates.

Sensitivity analyses were conducted by fitting generalised additive models to account for potential non-linearity of trends. All analyses were conducted with the SAS System for Windows, version 9.4 (SAS Institute). Models were fitted using the GENMOD and GAM procedures.

### Economic analysis

Postal testing kit data were combined with EPR data to estimate the total number of episodes per week (including those with negative postal tests and no clinic attendance). For estimating the difference in the mean number of episodes per week (1) negative postal test episodes were assigned to weeks pro rata with asymptomatic episodes that included clinic attendance and (2) the combined postintervention analysis used data for the first 43 weeks only. Episode costs were estimated using unit costs of diagnostic tests provided by the intervention site, and postal kit tests and staff time from the literature[12] inflated to 2021

values using a UK government gross domestic product deflator.[27] Treatment costs were from the British National Formulary[28] (online supplemental table S2). The cost of unreturned postal kits was allocated to episodes including a postal test result. CIs for differences in the number of episodes and cost per episode were calculated using the normal approximation method.

### Patient and public involvement

Three members of the public who had used the intervention site services as patients were involved in reviewing the proposed outcome measures and informed the study design.

### RESULTS

In the EHR intervention site data, 48776 attendances for females and 34413 for males were recorded during the study period, representing 32482 and 22073 episodes of care involving a clinic attendance, and 29573 and 19083 patients, respectively (table 2). Patients were symptomatic

**Table 2** Summary of population characteristics and outcomes by site, sex and time period based on EPR data

| | Intervention site | | Control site | |
| --- | --- | --- | --- | --- |
| | **Pre** | **Post** | **Pre** | **Post** |
| Males | | | | |
| Total attendances, n | 17626 | 16787 | 11920 | 12085 |
| Total episodes of care, n | 11445 | 10628 | 7946 | 8021 |
| Total patients, n | 9932 | 9151 | 6271 | 6335 |
| Symptomatic attendances, n (%) | 7307 (41.5%) | 7084 (42.2%) | 4735 (39.7%) | 4556 (37.7%) |
| Complex attendances, n (%) | 9869 (56.0%) | 9259 (55.2)% | 4458 (37.4%) | 4940 (40.9%) |
| Ethnic minority attendances, n (%) | 2834 (16.1%) | 3025 (18.0%) | 7244 (60.8%) | 7311 (60.5%) |
| MSM attendances, n(%) | 5300 (30.1%) | 5418 (32.3%) | 2529 (21.2%) | 2849 (23.6%) |
| Mean age, years | 30.2 | 30.8 | 34.9 | 35.1 |
| Urethral GC swabs per consultation | 0.18 | 0.11 | 0.08 | 0.07 |
| Median time-to-notification | 10.90 | 6.73 | 4.51 | 4.95 |
| Examinations per symptomatic attendance | 0.76 | 0.67 | 0.64 | 0.60 |
| Follow-up attendances per episode | 0.40 | 0.36 | 0.50 | 0.37 |
| Females | | | | |
| Total attendances | 28487 | 20289 | 20931 | 16910 |
| Total episodes of care | 18616 | 13866 | 13971 | 11660 |
| Total patients | 16779 | 12794 | 11799 | 9902 |
| Symptomatic attendances | 6312 (22.2%) | 4929 (24.3%) | 6860 (32.8%) | 5561 (32.9%) |
| Complex attendances | 26022 (91.3%) | 18173 (89.6%) | 12328 (58.9%) | 11221 (66.4%) |
| Ethnic minority attendances | 3979 (14.0%) | 3067 (15.1%) | 12647 (60.4%) | 10107 (59.8%) |
| Mean age | 25.1 | 25.8 | 29.8 | 30.4 |
| Cervical GC swabs per consultation | 0.20 | 0.04 | 0.03 | 0.03 |
| Median time-to-notification (median, IQR) | 10.58 | 3.52 | 4.90 | 5.32 |
| Examinations per symptomatic attendance | 0.73 | 0.70 | 0.58 | 0.60 |
| Follow-up attendances per episode | 0.36 | 0.34 | 0.31 | 0.23 |

EPR, electronic patient record; GC, gonorrhoea culture; MSM, men who have sex with men.

in just over 20% of female attendances and over 40% of male attendances. About 90% of female and 55% of male attendances were complex. Just over 30% of male attendances were by MSM.

## Males

For the rate of GC swabs, there was strong evidence of an adjusted step-increase for the intervention site relative to the control site (+89.1%, 95% CI +37.1% to +160.6%, p<0.001) (table 3 and figure 1A). However, this was not observed in the sensitivity analysis allowing for non-linear trends (−16.6%, 95% CI −30.1% to −0.5%, p<0.001, online supplemental table S3 and figure S4A). This was followed by strong evidence of an adjusted downward change in post-intervention trend of −3.2% per week (95% CI −4.3% to −2.1%, p<0.001). The long-term result of these two effects was an overall decrease from 35 to 50 swabs per week, pre-intervention, to below 10 per week at the end of the study period, translating to 849 swabs avoided over the postintervention period.

Time-to-notification increased by an estimated 3.6 days (95% CI 1.7 to 5.5 days, p<0.001) at the time of the intervention, relative to controls, and a similar increase was observed in the sensitivity analysis. However, this was followed by an overall long-term decrease of −0.2 days of notification time per week (95% CI −0.3 to −0.2 days, p<0.001) through the postintervention period. That is, the preintervention weekly median of around 8 to 9 days dropped to around 2 days after the intervention had been in place for a year (figure 1B, online supplemental figure S4B).

We found no evidence of a meaningful change in rates of examinations or follow-up attendances associated with the intervention (table 3, figure 1C,D, online supplemental figures S4C and S4D).

## Females

For females, there was evidence of a decrease in the rate of GC swabs: −40.8% (95% CI −61.6% to −8.8%, p=0.02) at the time of intervention, adjusted for control changes (table 3, figure 2A). This was followed by a decrease in trend through the postintervention period, with an adjusted change of −6.1% per week (95% CI −7.8% to −4.5%, p<0.001). These changes represent a decrease from an estimated 0.22 swabs per consultation (over 30 swabs per week) immediately before the intervention to 0.14 immediately after (20 to 25 per week) and down to 0.01 at the end of the study period (less than five per week). Over the 43-week postintervention period, an estimated 1542 swabs were avoided.

For time-to-notification, there was some evidence of a decrease of 2.1 days (95% CI −4.5 to 0.3 days, p=0.08, figure 2B) at the time of the intervention, adjusted for the control group. There was stronger evidence of a downward change in trend, estimated at −0.1 days per week (95% CI −0.20 to −0.0 days, p=0.01) over the postintervention period. These results were confirmed by the sensitivity analyses (online supplemental figure S5). To

**Table 3** Step change and slope change estimates from controlled interrupted time series models by outcome and sex

| Outcome | Change at time of intervention | | | Trend change following intervention | | |
| --- | --- | --- | --- | --- | --- | --- |
| | Intervention site | Control site | Intervention vs control | Intervention site | Control site | Intervention vs control |
| **Males—12 November 2018** | | | | | | |
| 1. Gonorrhoea culture swabs per consultation | +6.5% | −43.7% | +89.1% (+37.1%, +160.9%) | −3.6% | −0.3% | −3.2% (−4.3%, −2.1%) |
| 2. Time-to-notification | +2.2 days | +5.8 days | +3.6 days (+1.7, +5.5) | −0.19 days | +0.03 days | −0.2 days (−0.3 to −0.2) |
| 3. Examinations per symptomatic attendance | +3.6% | −1.6% | +5.4% (−7.5%, +20.0%) | −0.21% | −0.16% | −0.04% (−0.5%, +0.4%) |
| 4. Follow-up attendances per episode | −9.0% | −11.9% | +3.3% (−14.6%, +24.9%) | +0.23% | −0.001% | +0.30% (+0.31%, +0.96%) |
| **Females—29 May 2019** | | | | | | |
| 1. Gonorrhoea culture swabs per consultation | −38.7% | +3.6% | −40.8% (−61.6%, −8.8%) | −6.1% | −0.1% | −6.1% (−7.8%, −4.5%) |
| 2. Time-to-notification | −2.5 days | −0.4 days | −2.1 (−4.5, 0.3) days | −0.11 days | −0.0001 days | −0.1 (−0.2, −0.0) days |
| 3. Examinations per symptomatic attendance | −1.3% | −2.2% | +1.0% (−11.4%, +15.1%) | +0.09% | +0.03% | +0.1% (−0.4%, +0.5%) |
| 4. Follow-up attendances per episode | −8.2% | +2.7% | −10.6% (−27.6%, +10.3%) | −0.42% | +0.22% | −0.64% (−1.41%, +0.14%) |

Change estimates are shown for the intervention and control sites and for the relative change for intervention site compared with the control site. All estimates are shown as percentage changes, except for time-to-notification where change estimates are given in days.
Note: Results for outcome 5 (staff capacity) reported separately in the text.

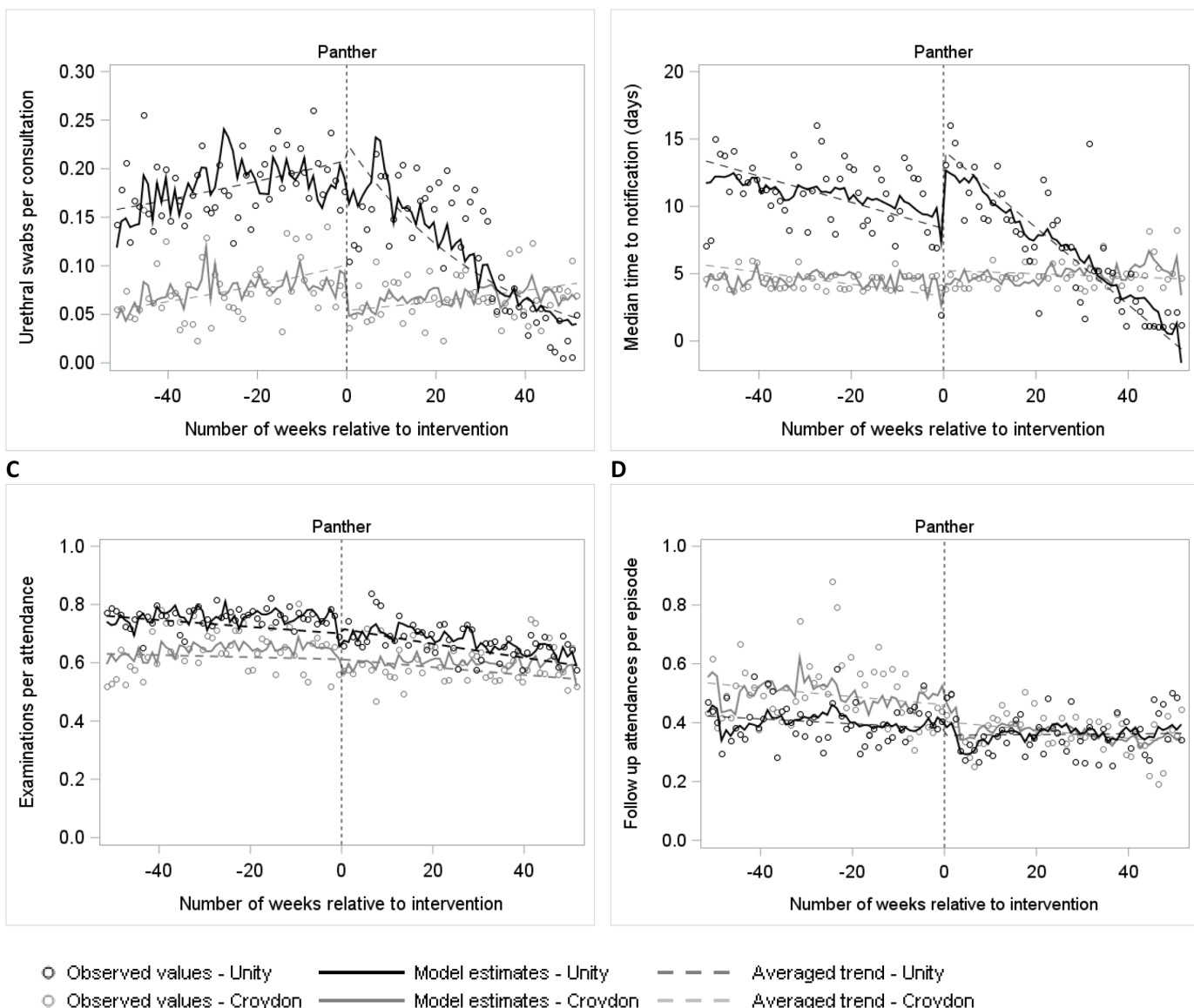

**Figure 1** Modelled outcome estimates for males. 'Panther' indicates the intervention date representing the first week the Panther system was implemented for the male pathway: 12 November 2018. (A) Gonorrhoea culture swabs (urethral) per consultation, (B) median time-to-notification, (C) examinations per symptomatic attendance, (D) follow-up attendances per episode.

illustrate, the estimated median time-to-notification was 8 to 9 days just before the intervention, but a year later notification time was around 1 day.

For rates of examinations and follow-up visits, we saw no evidence of intervention-related change (table 3, figure 2C,D).

### Staff capacity

The main analysis of staff capacity showed evidence of a trend change at the time of the male intervention (−1.1% per week, 95% CI −1.7% to −0.5%, p<0.001) and a step change at the time of the female intervention (+14.3%, 95% CI +3.4% to +26.3%, p=0.009) (figure 3). However, the sensitivity analysis showed step changes in the

opposite direction to the main analysis (online supplemental figure S6), suggesting inconclusive evidence of change.

### Episodes and costs

Overall, the intervention site experienced a substantial increase in the weekly number of asymptomatic negative episodes managed via postal test kits, particularly for men, while both asymptomatic negative episodes seen in the clinic and symptomatic episodes decreased (table 4). The mean cost per symptomatic episode increased by 9.2% to £69.04, while this was outweighed by a decrease of 13.5% to £26.23 for costs per asymptomatic episode, resulting in a combined decrease of 7.5%. The total cost

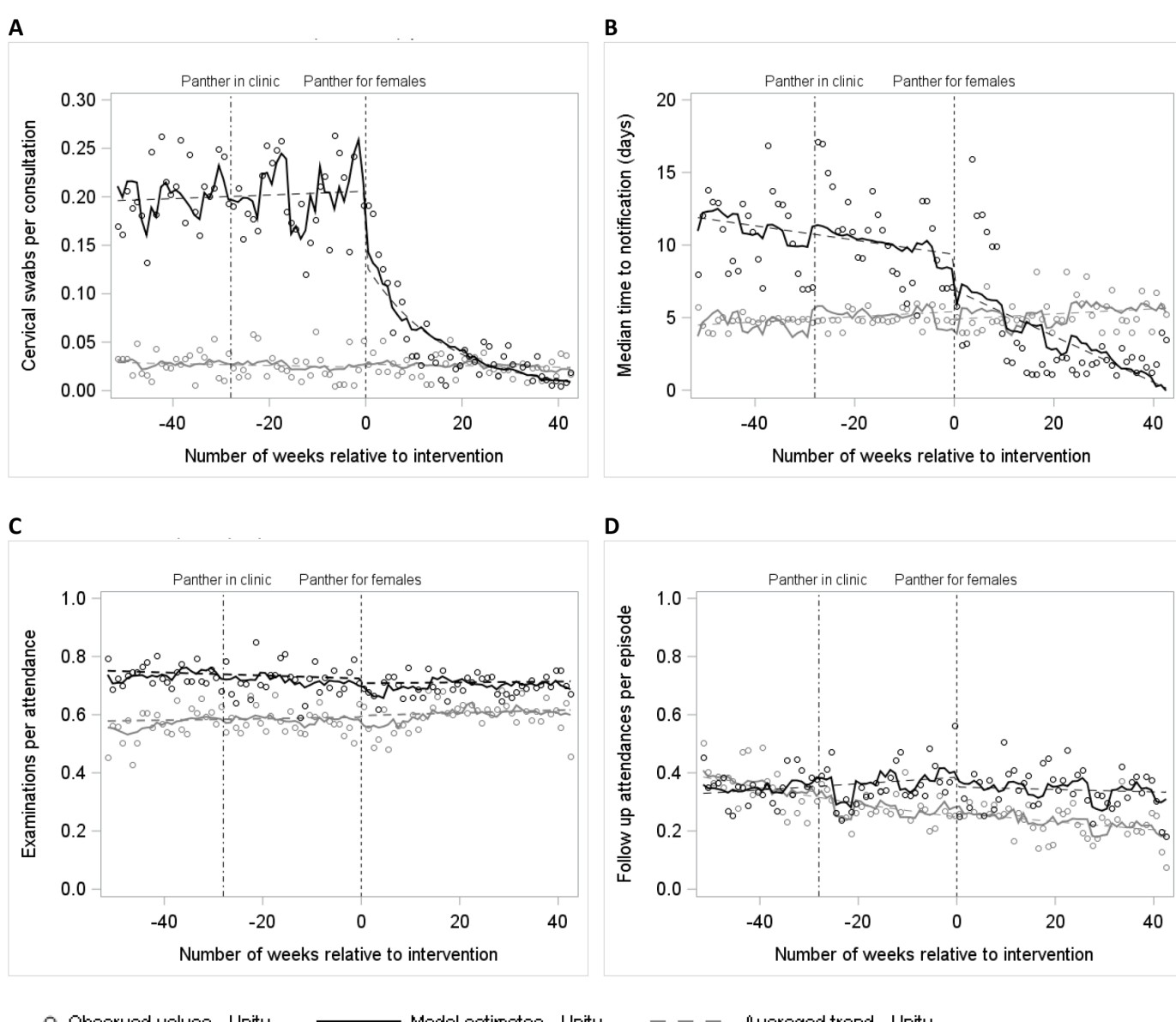

**Figure 2** Modelled outcome estimates for females. 'Panther for females' indicates the intervention date representing the first week the Panther system was implemented for the female pathway: 29 May 2019. (A) Gonorrhoea culture swabs (cervical) per consultation, (B) median time-to-notification, (C) examinations per symptomatic attendance, (D) follow-up attendances per episode.

per week decreased by 4.7%, largely due to the reduction in both the number and cost of episodes for asymptomatic females who attended the clinic.

## DISCUSSION

We have quantitatively evaluated the impact of a first-of-its-kind rapid STI testing system on service delivery in an integrated SHS. Previous NPT assessments have taken a mathematical modelling approach.[11–13] The only other study directly assessing a chlamydia and gonorrhoea NPT approach in practice related to a rapid testing service model for asymptomatic patients without contraception provision.[29] This is the first study to quantify the effect of rapid chlamydia and gonorrhoea NPT on GC swabs,

time-to-notification, examinations, follow-up visits, staff capacity and costs.

The substantial long-term postintervention decrease in the rate at which gonorrhoea swabs were sent for culture, for both males and females, was expected to some extent since patients with negative rapid tests in the new pathway avoided the need for cultures. Adams *et al*[11] identified reduced GCs as a key part of NPT-related cost reduction, although there has been no direct or simulated assessment of expected change in the number of cultures.

The trajectory of the decline in GC swab rates following the intervention differed between males and females. The sensitivity analysis capturing non-linear trends suggested substantial decreases for males began more

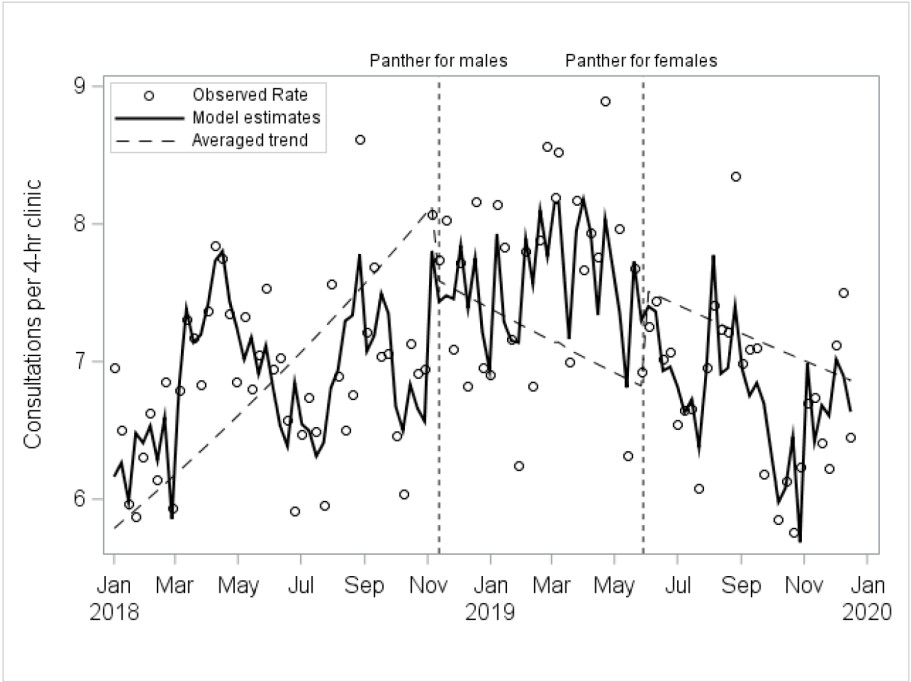

**Figure 3** Modelled estimates of staff capacity for males and females combined.

than 6 months after the intervention, with the lowest rates at 1-year post-intervention (online supplemental Figure S4A). In contrast, rates for females appeared to respond to the intervention almost immediately and stabilise at a much lower level within about 6 months (online supplemental Figure S5A). The differing implementation timeframes may reflect several barriers to implementation experienced with the initial rollout for males, including providing training to a large group staff with varying timetables exacerbated by understaffing and budget cuts; variable application of eligibility criteria for the new service and iterative revision of the new system and pathway.[30] There may also have been some just-in-case culture testing in the early stages until staff confidence in the system was established. With these issues largely resolved when the system was implemented for females, the transition appeared both smoother and faster, and this concurs with staff experience.

The rate of gonorrhoea swabs at the control site was relatively low throughout the period due to a conservative approach, appropriate to local prevalence, in which samples for cultures were only taken for NAAT-positive patients or those with high likelihood of infection. In contrast, standard practice at the intervention site in the pre-intervention period was to take cultures from all symptomatic patients with symptoms and/or signs potentially consistent with gonorrhoea and from potential contacts in addition to a NAAT as recommended in national guidelines.[31]

We estimated that median time-to-notification decreased from more than a week down to 1 or 2 days over the postintervention period. However, given that it was not possible to separate out all rapid test results (eg, notifications labelled 'all negative') and that we estimated

real time rather than working days, the median time was likely lower, particularly for positive results. This is broadly consistent with findings from Whitlock *et al*[29] who reported an average time-to-notification of 0.27 days for a new rapid NAAT testing service compared with 8.95 days for an off-site testing service for symptomatic patients.

The temporary increase in median time-to-notification for males after the intervention may result from the implementation challenges outlined above[30] in addition to a clinician-reported backlog in the early stages of transitioning to the new system. Once again, for males, the transition appeared to take place over the full post-intervention period, while the equivalent period for females appeared faster with the lowest post-intervention sensitivity estimates occurring 21 weeks after the new system was implemented (online supplemental figures S4B and S5B).

We observed no clear evidence of intervention-related changes in rates of examinations, follow-up visits or staff capacity. All three were necessarily constructed from combinations of variables as there was no dedicated data field for each in the data. Although we did not detect a positive change, it is important to note that there was no evidence of a deleterious impact of the rapid testing service on any of these outcomes.

Staff capacity showed some evidence of intervention-related change, although the rate of patients seen per 4 hour clinic was at similar levels at the end of the study period as at the start. For asymptomatic patients, the provision of postal testing kits reduced the need for clinic attendance among those testing negative both for males and for females who did not have contraception needs. This combined with the introduction of shorter appointments more than likely increased staff capacity for this

**Table 4** Intervention site preintervention and postintervention estimates of mean number of episodes per week, mean cost per episode and mean cost per week

| | Male | | | | | Female | | | | | Total | | | | |
|---|---|---|---|---|---|---|---|---|---|---|---|---|---|---|---|
| | Pre* | Post* | % change | 95%CI | | Pre* | Post† | % change | 95%CI | | Pre* | Post† | % change | 95%CI | |
| *Mean number per week* | | | | | | | | | | | | | | | |
| Asymptomatic | 190.2 | 223.1 | 17.3 | 9.5 | 25.1 | 356.2 | 350.7 | −1.5 | −7.9 | 4.9 | 546.3 | 573.4 | 5.0 | 0.0 | 9.9 |
| Postal negatives | 70.5 | 111.5 | 58.2 | 48.7 | 67.7 | 96.3 | 124.9 | 29.7 | 22.3 | 37.0 | 166.8 | 236.2 | 41.6 | 35.7 | 47.4 |
| Other‡ | 119.6 | 111.5 | −6.8 | −13.7 | 0.2 | 259.9 | 225.9 | −13.1 | −19.2 | −7.0 | 379.5 | 337.2 | −11.1 | −15.8 | −6.5 |
| Symptomatic | 92.7 | 85.0 | −8.3 | −13.9 | −2.7 | 84.4 | 77.8 | −7.8 | −14.6 | −1.1 | 176.7 | 163.4 | −7.5 | −11.8 | −3.2 |
| Total | 282.8 | 308.0 | 8.9 | 2.6 | 15.2 | 440.2 | 429.0 | −2.5 | −8.7 | 3.6 | 723.0 | 736.8 | 1.9 | −2.5 | 6.3 |
| *Cost per episode (£)* | | | | | | | | | | | | | | | |
| Asymptomatic | 36.47 | 30.92 | −15.2 | −19.1 | −11.3 | 27.04 | 24.23 | −10.4 | −13.3 | −7.5 | 30.31 | 26.23 | −13.5 | −15.9 | −11.0 |
| Symptomatic | 63.09 | 69.56 | 10.3 | 6.7 | 13.8 | 63.36 | 67.65 | 6.8 | 4.3 | 9.2 | 63.22 | 69.04 | 9.2 | 6.9 | 11.5 |
| Total | 45.19 | 41.58 | −8.0 | −10.8 | −5.2 | 33.98 | 32.14 | −5.4 | −7.7 | −3.1 | 38.36 | 35.47 | −7.5 | −9.3 | −5.7 |
| *Cost per week (£)* | | | | | | | | | | | | | | | |
| Resource | | | | | | | | | | | | | | | |
| Postal kit | 382 | 592 | 55.0 | 45.9 | 64.1 | 629 | 848 | 34.8 | 27.4 | 42.2 | 1010 | 1437 | 42.3 | 36.5 | 48.1 |
| In clinic diagnostic test | 1962 | 1886 | −3.9 | −9.8 | 2.1 | 1452 | 1213 | −16.5 | −22.9 | −10.1 | 3413 | 3155 | −7.6 | −11.9 | −3.3 |
| Consultation staff time | 7497 | 7349 | −2.0 | −7.3 | 3.4 | 9396 | 8583 | −8.7 | −15.0 | −2.3 | 16893 | 15959 | −5.5 | −9.5 | −1.5 |
| Treatment | 3024 | 2896 | −4.2 | −13.1 | 4.6 | 3534 | 3085 | −12.7 | −20.3 | −5.1 | 6558 | 6014 | −8.3 | −14.4 | −2.2 |
| Symptom status | | | | | | | | | | | | | | | |
| Asymptomatic | 6949 | 6883 | −1.0 | −8.5 | 6.6 | 9673 | 8448 | −12.7 | −18.9 | −6.4 | 16622 | 15392 | −7.4 | −12.3 | −2.5 |
| Symptomatic | 5915 | 5840 | −1.3 | −7.5 | 5.0 | 5338 | 5280 | −1.1 | −8.7 | 6.6 | 11253 | 11174 | −0.7 | −5.5 | 4.0 |
| Total | 12865 | 12723 | −1.1 | −6.7 | 4.5 | 15010 | 13728 | −8.5 | −14.4 | −2.6 | 27875 | 26565 | −4.7 | −8.6 | −0.8 |

*Based on 52 week period.
†Based on 43 week period.
‡Includes positive postal test kits.

subgroup. Both also reduced the queuing time for walk-in clinics. Conversely, the reduced asymptomatic attendances meant that case-mix in the walk-in clinics became more demanding, with patients more likely to be symptomatic and/or complex,[30] which may explain the lack of observed improvement in staff capacity during clinics. The lack of evidence for a capacity decrease through the implementation period despite a more demanding patient group and the growing numbers of asymptomatic patients being tested both suggest increased capacity of the SHS overall.

The change in management of asymptomatic clinical attendances, supported by the existing postal testing kit system, was a key component of the overall cost reduction following the introduction of the Panther technology, with decreases in both mean cost per asymptomatic episode (13.5%) and weekly asymptomatic costs (7.4%). Although the cost of symptomatic episodes increased, consistent with the reported increase in complexity of symptomatic patients in clinic, this was counteracted by a reduction in the number of weekly symptomatic attendances.

## Strengths and limitations

We conducted a prospective real-time evaluation of a large integrated rapid STI service. We

used a CITS framework with both a control site and confounder adjustment to estimate the effect of the intervention distinct from any background changes and independent of other time-varying factors. This was bolstered by using a relatively long time series with good temporal resolution. The robustness of our analysis was supported by both sites using the same EPR system and the general consensus between main and sensitivity analyses.

In light of the target trial framework for natural experiments,[32] our study was limited by being non-randomised, having only one control site, relying on the construction of certain outcomes from multiple variables, and the impact of the COVID-19 pandemic on the follow-up period for females. The unit costs were based on data provided by the intervention site and estimates from literature, and commissioners will need to assess their applicability to their locality.

## Implications and conclusions

Several studies have suggested that NPT benefits include earlier diagnosis and treatment, reduced risk of sequelae and onward transmission, reduction in unnecessary treatments, earlier partner notification and reduced anxiety.[10 29]

This quantitative assessment of the first UK implementation of rapid chlamydia and gonorrhoea testing within an integrated service revealed clear benefits, namely: reduced GC swabs and shortened time-to-notification. These improvements, while maintaining activity at a lower overall cost, suggest that the introduction of clinic-based rapid testing had the intended impact, and this is in line with previous NPT modelling studies.[10 11] The qualitative evaluation of this rapid STI service also reported that

patients valued faster results and avoiding unnecessary treatment, and that the better targeting of infection-specific treatment improved antimicrobial stewardship.[30] Although this was an evaluation of an integrated SHS providing contraception care in addition to testing, treatment and prevention services, it is likely that the findings would be applicable to SHSs that do not provide contraception care.

These results provide real-life evidence to support the benefits of a rapid testing service anticipated by modelling studies and strengthen the case for more widespread rollout in SHS.

**Author affiliations**
[1]National Institute for Health and Care Research, Applied Research Collaboration West (NIHR ARC West), University Hospitals Bristol and Weston NHS Foundation Trust, Bristol, UK
[2]Population Health Sciences, Bristol Medical School, University of Bristol, Bristol, UK
[3]Unity Sexual Health, University Hospitals Bristol and Weston NHS Foundation Trust, Bristol, UK
[4]Bristol Haematology and Oncology Centre, University Hospitals Bristol and Weston NHS Foundation Trust, Bristol, UK
[5]Croydon Sexual Health, Croydon University Hospital, Croydon, UK
[6]Southwest Regional Laboratory, UK Health Security Agency, North Bristol NHS Trust, Bristol, UK
[7]National Institute for Health and Care Research, Health Protection Research Unit in Behavioural Science and Evaluation (NIHR HPRU), University of Bristol, Bristol, UK

**Acknowledgements** The authors would like to thank Ed Hulse at Mill Systems for his indispensable assistance with the data extracts.

**Contributors** PJH, MTR, FDV and HM conceptualised the evaluation; MTR and FDV are quantitative evaluation leads; WH and HM are health economic evaluation leads; SRW, JJ, RM and MTR acquired the analysis datasets; SRW conducted the time series analysis with support from JJ, RM, MTR, PH and FDV; GM conducted the cost-effectiveness analysis with support from HM and WH; RG, MDC, MC, DP, PM, JS and JT advised on the study methodology, analysis and interpretation of results; SRW wrote the initial draft of the manuscript; all authors reviewed and edited the manuscript for content and approved the submission; SRW is the guarantor and takes responsibility for the overall content.

**Funding** This research was funded by the National Institute for Health Research (NIHR) Applied Research Collaboration West (ARC West) at University Hospitals Bristol and Weston NHS Foundation Trust (core NIHR infrastructure funded: NIHR200181). The views expressed are those of the authors and not necessarily those of the NIHR or the Department of Health and Social Care. FDV is partly funded by the NIHR School for Public Health Research.

**Competing interests** None declared.

**Patient and public involvement** Patients and/or the public were involved in the design, or conduct, or reporting, or dissemination plans of this research. Refer to the Methods section for further details.

**Patient consent for publication** Not applicable.

**Ethics approval** This study was approved by the Health Research Authority (South West) Research Ethics Committee, reference 18/SW/0090.

**Provenance and peer review** Not commissioned; externally peer reviewed.

**Data availability statement** Anonymised individual-level data for this study come from the electronic patient record system of the Unity Sexual Health and Croydon Sexual Health services (data controllers). Our data sharing agreement with the data controllers prohibits sharing data extracts outside of the University of Bristol research team. The data are available upon request from the data controllers.

of the translations (including but not limited to local regulations, clinical guidelines, terminology, drug names and drug dosages), and is not responsible for any error and/or omissions arising from translation and adaptation or otherwise.

**ORCID iDs**
Scott R Walter http://orcid.org/0000-0002-1898-6301
Ruta Margelyte http://orcid.org/0000-0002-7914-8037

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
