## [Reviewer comments · BMJ Open]

ARTICLE DETAILS

TITLE (PROVISIONAL)	The impact of rapid near-patient STI testing on service delivery outcomes in an integrated sexual health service in the United Kingdom: a controlled interrupted time series study
AUTHORS	Walter, Scott; Jackson, Joni; Myring, Gareth; Redaniel, Maria Theresa; Margelyte, Ruta; Gardiner, Rebecca; Clarke, Michael; Crofts, Megan; McLeod, Hugh; Hollingworth, William; Phillips, David; Muir, Peter; Steer, Jonathan; Turner, Jonathan; Horner, Paddy; De Vocht, Frank

VERSION 1 – REVIEW

REVIEWER	Wilson, Peter University College London Hospitals NHS Foundation Trust, Microbiology
REVIEW RETURNED	17-Jun-2022

GENERAL COMMENTS	The utility of near patient testing in STI diagnosis depends on various factors including cost, speed, quality control and staffing to perform the test. The greatest potential risk is misdiagnosis as the result of poor technique which does not arise in the laboratory where there is strict quality control. This study suggests a faster delivery of a result and reduced costs. Faster delivery facilitates a reduction in onward transmission. Quality control at point of care does not seem to have been addressed in this paper. The intervention was not randomised and truncated by Covid-19. The authors evaluated the impact of near patient testing using interrupted time series method. Culture for gonococci was only used for those with a positive NAAT result. Results for men were available after 4 hours but 48 hours for women. A different clinic was used as control. Time to notification in models was treated as a normal distribution but weekends and holidays may have affected it and is more likely to have been skewed. This seems to have been acknowledged by using median days instead of mean. There was said to be a large decrease in the number of GC cultures per consultation at the male control clinic in Croydon, with increased turnaround but this was not explained. Even though allowance was made in the analysis, this change at the control clinic may have affected the interpretation of the results of the intervention at the test clinic and needs to be checked by a statistical referee. The graphs suggest little change in control rates. There was also an effect due to postal testing but the timescales for this service are unclear. If the reduction in swabs and faster time to notification were correct, then the intervention would be worthy of more widespread implementation, but these issues need to be clarified.
--

REVIEWER	Kerdelmidis, Melissa Canterbury District Health Board, Planning & Funding
REVIEW RETURNED	23-Jun-2022

GENERAL COMMENTS	An interesting study. A few minor edits would improve. Please define an ethnic minority. The time to text message notification of test result is useful, but not everyone will have returned for a script – did you also have a measure of those who returned for a script or those who were lost to follow up? (Pls include if poss). Suggest a sentence as to the purpose of referring MSM to a health advisor. Was this for HIV testing, as that was not mentioned at all? Postal testing kits and their role should be mentioned a bit earlier, in the Introduction, they come up for the first time in the Discussion. Table 3 would benefit from a clearer explanation as to what this represents. If known or possible, it would be interesting to know how this service compares in terms of STI service times, next to others in the region. Was this new testing regime a lot faster than that used by others?
--

REVIEWER	Brennan-Krohn, Thea Boston Children's Hospital
REVIEW RETURNED	01-Jul-2022

GENERAL COMMENTS	General comments: The authors present data on the effects of implementation of a new rapid STI testing algorithm. Their study design allowed for prospective evaluation of an intervention site and included a control site. This is an important public health topic and the study overall was well-designed. In addition to some specific suggestions/questions below, my main concern is that, as seen in Figures 1A-B and 2A-B, the control site already had much lower rates of gonorrhoea swabs and time to notification during the pre-intervention period. I think this merits significant discussion in the text, including discussion of what the control site might have been doing differently than the test site before the intervention and how it was chosen as a control site. Furthermore, because the starting rates in the control site were so low, it might have been hard to detect much change in these rates, and therefore the fact that the rates of change were higher in the intervention group might be of less significance. It seems clear from looking at the data for the intervention site alone that the intervention appeared to result in a reduction in both of these variables, so I wonder whether the data might be more convincing if it were simply presented in terms of before-and-after data for the intervention group. In any case, this discrepancy between sites merits some discussion and explanation. Specific suggestions: Throughout: I would suggest referring to the sites as “control site” and “intervention site” or something to that effect, rather than Croydon and Unity, so that readers don't have to refer back to remember which is which. The authors refer several times to “postal testing kits”, but it is not clear to me what these are or how they are integrated into the algorithm. (Perhaps this is something that would be obvious to
---

	readers in the UK, but it is unfamiliar to me as a US reviewer and would presumably be unfamiliar to other international readers). Abstract: Page 2, line 46/47: Introduction: Page 4, line 30/31: Please define or give example(s) of near-patient testing. Page 4, lines 35-38: Is the idea that gonorrhoea cultures are reduced because NAAT is being used instead? I would clarify this, otherwise it may be interpreted as suggested that reduced gonorrhoea testing is the goal. Methods: Page 8, line 54/55 (and supplemental material): I'm curious about how the authors came up with the criteria for complex cases, particularly since 90% of female patients ended up meeting this criteria. For example, it isn't clear to me how the fact that a woman receives contraceptive care would make her presentation for STI testing more complex. I think some explanation would be helpful, and if these are standard criteria in this field then that could be stated. Page 9, line 10: How was "marked change over time" defined? Results: General: I think it is unnecessarily confusing to describe changes present at the time of intervention except in cases where the authors think that post-intervention results are related to pre-existing trends rather than to the intervention. I understand that including and analyzing this pre-intervention data is important in ensuring that changes were actually related to the intervention and not reflective of some pre-existing trend, but I don't think it needs to be automatically included in the text for each result. Table 2: I would suggest separating this into two tables, one for baseline population characteristics and a second one for outcomes. Discussion: Page 17, line 52 page 18, line 6: When the authors state that the case-mix in walk-in clinics became more complex, it isn't clear whether they're referring to specific changes in their clinics or to general trends.
--	---

REVIEWER	Liu, Jin Vertex Pharmaceuticals Incorporated
REVIEW RETURNED	03-Aug-2022

GENERAL COMMENTS	It is better to include the rationale of including all two-ways and three-ways interactions in the CITS approach. Is it driven by statistical metrics such as BIC or driven by answering specific research questions? Do the analysis results suggest these interactions statistically significant?
---

VERSION 1 – AUTHOR RESPONSE

Reviewer 1: Dr. Peter Wilson	
3. The utility of near patient testing in STI diagnosis depends on various factors including cost, speed, quality control and staffing to perform the test. The greatest potential risk is misdiagnosis as the result of poor technique which does not arise in the laboratory where there is strict quality control. This study suggests a faster delivery of a result and reduced costs. Faster delivery facilitates a reduction in onward transmission. Quality control at point of care does not seem to have been addressed in this paper.	In the period preceding the intervention, most patients took their own samples, so there was not necessarily a change in practice with the new system regarding taking samples. There is also evidence that self-taken samples result in comparable diagnostic accuracy compared to clinician taken samples (see: https://doi.org/10.1093/cid/ciaa1266 and https://doi.org/10.1093/cid/ciaa1546)
4. The intervention was not randomised and truncated by Covid-19. The authors evaluated the impact of near patient testing using interrupted time series method. Culture for gonococci was only used for those with a positive NAAT result. Results for men were available after 4 hours but 48 hours for women. A different clinic was used as control. Time to notification in models was treated as a normal distribution but weekends and holidays may have affected it and is more likely to have been skewed. This seems to have been acknowledged by using median days instead of mean.	As noted by the reviewer, the median weekly time to notification was used to account for the skewed nature of the time interval data, and this would also be resilient to the effect of public holidays. This was confirmed when we reran the models adjusting for weeks in which there were public holidays and observed a negligible effect on the main estimates for intervention-related change. In practical terms, the turnaround time for all patients using the rapid system was around four hours. For administrative purposes a 48 hour period was allowed for results to be reported in the female pathway, however, we specified a 4 hour turnaround time within the male pathway when managing men with symptoms of urethritis.
5. There was said to be a large decrease in the number of GC cultures per consultation at the male control clinic in Croydon, with increased turnaround but this was not explained. Even though allowance was made in the analysis, this change at the control clinic may have affected the interpretation of the results of the intervention at the test clinic and needs to be checked by a statistical referee. The graphs suggest little change in control rates. There was also an effect due to postal testing but the timescales for this service are unclear. If the reduction in swabs and faster time to notification were correct, then the intervention would be worthy of more widespread implementation, but these issues need to be clarified.	The decrease in rates of GC culture for the control site resulted in an amplified relative step change for the intervention site: a relative 89% increase compared to 6.5% actual increase. This upward step change was not supported by the sensitivity analysis so was not discussed in depth. We also refrained from discussing the control site decrease as the key feature of this part of the analysis was the clear long-term decrease for the intervention site. The purpose of modelling these outcomes is to provide insights related to specific research questions which may identify results that are not otherwise apparent from looking at the graphs. Different practices for undertaking GC cultures were in place at each centre. The intervention site followed national guidelines. Asymptomatic

	patients were not routinely cultured prior to the intervention unless they were a contact of someone with a positive GC result. In addition, GC cultures are not performed via postal testing kits for various reasons including test performance. Thus, increasing postal NAAT testing for other infections at the intervention site on asymptomatic patients over time is very unlikely to have affected GC culture rates. We have included a paragraph in the discussion about this (paragraph 4).
Reviewer 2. Dr. Melissa Kerdelmelidis	
6. Please define an ethnic minority	This is self-reported at patient registration using the nationally-defined categories that are used within the patient record system. Categories included in our definition were: African, Caribbean, any other black background, white and black Caribbean, white and black African; Indian, Pakistani, Bangladeshi, any other Asian background, white and Asian; Chinese; Any other mixed background; Any other ethnic group. This has now been added to the final page of the supplementary material.
7. The time to text message notification of test result is useful, but not everyone will have returned for a script – did you also have a measure of those who returned for a script or those who were lost to follow up? (Pls include if poss).	This is an important research question, albeit nontrivial to answer. Unfortunately, we do not have complete data on this readily available from the EPRs. Researchers at the intervention site have undertaken a separate analysis over a 2 month period before and after the introduction of the rapid STI service which indicated time to treatment for chlamydia decreased from 8.7 to 5.1 days ($p < 0.001$). This is the subject of planned future research building on this conference poster: https://sti.bmj.com/content/7/Suppl_1/A148.3
8. Suggest a sentence as to the purpose of referring MSM to a health advisor. Was this for HIV testing, as that was not mentioned at all?	The sentence has now been augmented as follows: “Men who have sex with men (MSM) were referred to a health advisor for health promotion, including discussion about testing for HIV and other STIs, and safer sex practices.”
9. Postal testing kits and their role should be mentioned a bit earlier, in the Introduction, they come up for the first time in the Discussion.	We have added some text to section 2.1 (first paragraph) providing more detail about the postal testing kit component of the service: “In addition to in-clinic services, self-testing kits for chlamydia, gonorrhoea, syphilis and HIV ordered online by patients are provided by post. This postal testing kit service was provided by the intervention site for asymptomatic patients through its dedicated

	website and used the same NAAT testing platform as the rapid STI service. This was in place prior to the intervention and was increasingly used throughout the study period.”
10. Table 3 would benefit from a clearer explanation as to what this represents.	Additional description has been added to caption of Table 3 to clarify the information it presents.
11. If known or possible, it would be interesting to know how this service compares in terms of STI service times, next to others in the region. Was this new testing regime a lot faster than that used by others?	The control site was used as an indication of turnaround times in the absence of a rapid testing service, although this was outside the immediate region. We do not have comparable data from any other clinics in South West England.
Reviewer 3. Dr. Thea Brennan-Krohn	
12. The authors present data on the effects of implementation of a new rapid STI testing algorithm. Their study design allowed for prospective evaluation of an intervention site and included a control site. This is an important public health topic and the study overall was well-designed. In addition to some specific suggestions/questions below, my main concern is that, as seen in Figures 1A-B and 2A-B, the control site already had much lower rates of gonorrhoea swabs and time to notification during the pre-intervention period. I think this merits significant discussion in the text, including discussion of what the control site might have been doing differently than the test site before the intervention and how it was chosen as a control site. Furthermore, because the starting rates in the control site were so low, it might have been hard to detect much change in these rates, and therefore the fact that the rates of change were higher in the intervention group might be of less significance. It seems clear from looking at the data for the intervention site alone that the intervention appeared to result in a reduction in both of these variables, so I wonder whether the data might be more convincing if it were simply presented in terms of before-and-after data for the intervention group. In any case, this discrepancy between sites merits some discussion and explanation.	The control site was chosen since it was also a NHS-run integrated sexual health service with a similar level of patient throughput, that uses the same electronic patient record system. Although the rates for the control site are considerably lower than for the intervention site, including a control series in the analysis is important for establishing intervention-related changes as distinct from any background changes. We feel that presenting those control-related results is important in providing a complete picture of our evaluation. A paragraph has been added to the discussion (paragraph 4) addressing the difference in baseline rates of gonorrhoea cultures (see also comment 5 above):
13. Throughout: I would suggest referring to the sites as “control site” and “intervention site” or something to that effect, rather than Croydon and Unity, so that readers don’t have to refer back to remember which is which.	The text has been updated accordingly.

14. The authors refer several times to “postal testing kits”, but it is not clear to me what these are or how they are integrated into the algorithm. (Perhaps this is something that would be obvious to readers in the UK, but it is unfamiliar to me as a US reviewer and would presumably be unfamiliar to other international readers). Including, abstract: Page 2, line 46/47	In response to this comment and comment 9 above we have added further description of the postal testing kit service to section 2.1. Availability of online gonorrhoea and chlamydia NAAT tests and syphilis and HIV testing for asymptomatic patients using postal kits is now widespread for patients accessing sexual health services in the UK.
15. Introduction: Page 4, line 30/31: Please define or give example(s) of near-patient testing.	We have provided a definition at the first mention of the term.
16. Page 4, lines 35-38: Is the idea that gonorrhoea cultures are reduced because NAAT is being used instead? I would clarify this, otherwise it may be interpreted as suggested that reduced gonorrhoea testing is the goal.	The reviewer is correct in that only those patients with NAAT positivity are cultured or if gonococcal treatment was administered prior to the NAAT result., which reduces costs and unnecessary testing. The text has been changed to: “...reduction in the need for gonorrhoea cultures...”
17. Methods: Page 8, line 54/55 (and supplemental material): I’m curious about how the authors came up with the criteria for complex cases, particularly since 90% of female patients ended up meeting this criteria. For example, it isn’t clear to me how the fact that a woman receives contraceptive care would make her presentation for STI testing more complex. I think some explanation would be helpful, and if these are standard criteria in this field then that could be stated.	In the UK the majority of SHS are now integrated, meaning they also provide a full range of contraceptive services (now explained in the introduction). The definition of complexity for both males and females was based on a previous modelling study (Mohiuddin et al., 2020 – ref 12). In both that study and in ours, the definition of ‘complex’ is to some extent a proxy for consultation time and contraception-related consultations tend to be longer and more involved. A high proportion of female attendances also tend to be contraception-related. We have added a sentence to the methods accordingly (section 2.5 paragraph 4): “This is based on the definition used by Mohiuddin et al.[12] designed to identify patients requiring longer and/or more involved consultations.”
18. Page 9, line 10: How was “marked change over time” defined?	This part of the sentence is redundant since we applied the sensitivity analyses to all outcomes as shown in the supplement (Figures S4-S6). The sentence now reads: “Sensitivity analyses were conducted by fitting generalised additive models to account for potential non-linearity of trends”
19. Results: General: I think it is unnecessarily confusing to describe changes present at the time of intervention except in cases where the authors think that post-intervention results are related to pre-existing trends rather than to the intervention. I understand that including and analyzing this pre-intervention data is important in ensuring that changes	We have removed results that present the intervention and control changes separately, and instead focus on the relative change as is the purpose for controlled interrupted time series analysis. It is necessary to include step change which involves comparing the pre and post outcome levels at the time of the intervention as this captures short term impact of the intervention. The takeaway from

were actually related to the intervention and not reflective of some pre-existing trend, but I don't think it needs to be automatically included in the text for each result.	the results is more about the long-term impact but we feel it is important to present the whole change trajectory.
20. Table 2: I would suggest separating this into two tables, one for baseline population characteristics and a second one for outcomes.	We are happy to split these components into two separate tables but defer to the editors on this decision.
21. Discussion: Page 17, line 52 page 18, line 6: When the authors state that the case-mix in walk-in clinics became more complex, it isn't clear whether they're referring to specific changes in their clinics or to general trends.	In addition to providing more detail about the postal testing kits (see comments 9 and 14) we have clarified this section of the discussion (paragraph 8): "For asymptomatic patients, the provision of postal testing kits reduced the need for clinic attendance among those testing negative both for males and for females if they did not have contraception needs. This combined with the introduction of shorter appointments more than likely increased staff capacity for this subgroup. Both also reduced the queueing time for walk-in clinics. Conversely, the reduced asymptomatic attendances meant that case-mix in the walk-in clinics became more demanding, with patients more likely to be symptomatic and/or complex, which may explain the lack of observed improvement in staff capacity during clinics."
Reviewer 4. Dr. Jin Liu	
22. It is better to include the rationale of including all two-ways and three-ways interactions in the CITS approach. Is it driven by statistical metrics such as BIC or driven by answering specific research questions? Do the analysis results suggest these interactions statistically significant?	The interactions are a key part of estimating the changes in the intervention group relative to the control group, as described in the methods section and illustrated in Figure S3. Our interest in the three-way interaction time x group x period (relative slope change) necessitates including all two-way interactions.

VERSION 2 – REVIEW

REVIEWER	Wilson, Peter University College London Hospitals NHS Foundation Trust, Microbiology
REVIEW RETURNED	24-Oct-2022

GENERAL COMMENTS	The authors have answered most of the comments. However the point regarding laboratory quality control has not because the authors addressed the specimen collection not the performance of the test. Quality control protocols at the point of care need to be described whether in a satellite laboratory or elsewhere. It mentions quality is maintained only.
---

REVIEWER	Liu, Jin Vertex Pharmaceuticals Incorporated
-----------------	---

REVIEW RETURNED	31-Oct-2022
GENERAL COMMENTS	The authors have addressed the comments in the previous version.

VERSION 2 – AUTHOR RESPONSE

Response to reviewer comments: bmjopen-2022-064664.R2

COMMENT	RESPONSE
Reviewer 1: Dr. Peter Wilson	
1. The authors have answered most of the comments. However the point regarding laboratory quality control has not because the authors addressed the specimen collection not the performance of the test. Quality control protocols at the point of care need to be described whether in a satellite laboratory or elsewhere. It mentions quality is maintained only.	We have added a subsection to the Methods (see section 2.2.1) providing details of the rapid STI testing including aspects of quality control. We have also provided additional detail in the supplementary material (p.2).